# Double $bla_{KPC-2}$ copies quadrupled minimum inhibitory concentration of ceftazidime–avibactam in hospital-derived *Klebsiella pneumoniae*

Dakang Hu,[1] Shijie Wang,[2] Mengqiao Xu,[1] Jin Zhang,[1] Xinhua Luo,[1] Wei Zhou,[3] Qinfei Ma,[1] Xiaobo Ma[4]

**ABSTRACT**  To illustrate the genomic and drug resistance traits of the *Klebsiella pneumoniae* Kpn_XM9, which harbors a transposon (Tn) As1 and was barely susceptible to ceftazidime–avibactam (CZA). Whole-genome sequencing, gene deletion, antimicrobial susceptibility, and conjugation tests were carried out to illustrate the traits of Kpn_XM9. As confirmed by whole-genome sequencing, the Kpn_XM9 harbored a 5,523,536 bp chromosome and five plasmids with lengths being 128,129, 196,512, 84,812, 43,695, and 5,596 bp, respectively. Plasmid p1_Kpn_XM9 (128,219 bp) contained four resistance genes, $bla_{CTX-M-65}$, $bla_{TEM-1B}$, $rmt$B, and two copies of $bla_{KPC-2}$. Genes $bla_{KPC-2}$ were bracketed by ISKpn17 and ISKpn16 within a new composite Tn3-like TnAs1. The two tandem repeats, positioned opposite each other, were spaced 93,447 bp apart in p1_Kpn_XM9. Kpn_XM9 belonged to K64 and sequence type (ST) 11. The Kpn_XM9 was resistant to amikacin, aztreonam, ticarcillin/clavulanic acid, piperacillin/tazobactam, ceftazidime, cefepime, imipenem, meropenem, tobramycin, ciprofloxacin, levofloxacin, doxycycline, minocycline, tigecycline, colistin, and trimethoprim/sulfamethoxazole; it was barely susceptible to CZA with a minimum inhibitory concentration of 8/4 µg/mL, which declined to 2/4 µg/mL after a 18,555 bp nucleotide was knocked out and one copy of $bla_{KPC-2}$ was sustained on p1_Kpn_XM9. Kpn_XM9 had virulence genes encoding Types 1 and 3 fimbriae, four siderophores, and capsular polysaccharide anchoring protein but no genes upregulating capsular polysaccharide synthesis. The Kpn_XM9 presented a classical phenotype with extreme drug resistance. The emergence of double copies of $bla_{KPC-2}$ in a single plasmid from the predominant ST11 *K. pneumoniae* represents a new therapeutic challenge.

**IMPORTANCE**  With the wide use of ceftazidime–avibactam against carbapenem-resistant organisms, its resistance is increasingly documented; among the corresponding resistance mechanisms, mutations of $bla_{KPC-2}$ or $bla_{KPC-3}$ into other subtypes are dominant to date. However, more copies of $bla_{KPC-2}$ may also greatly increase the minimum inhibitory concentration of ceftazidime–avibactam, which could be conferred by transposon As1 and insertion sequence 26 and should be of concern.

**KEYWORDS**  *Klebsiella pneumoniae*, ceftazidime–avibactam, drug resistance, transposon, KPC-2

Carbapenems are considered antibiotics of choice against multidrug-resistant and extended-spectrum β-lactamase-producing strains, but the global increase of carbapenem-resistant *Enterobacterales* (CRE) is compromising their efficacy in therapy (1). Carbapenemases frequently encoded by genes located on transferable elements and isolates of *Escherichia coli*, *Klebsiella pneumoniae*, and *Enterobacter* spp., carrying multiple

Address correspondence to Qinfei Ma, 13957687070@163.com, or Xiaobo Ma, maxiaoboxm@126.com.

Dakang Hu, Shijie Wang, and Mengqiao Xu contributed equally to this article. The author order was determined both alphabetically and in order of increasing seniority.

The authors declare no conflict of interest.

See the funding table on p. 9.

carbapenemase-encoding genes on plasmids of different incompatibility (Inc) groups, have been reported (2).

*K. pneumoniae* is one of the most important opportunistic pathogens causing severe nosocomial infections such as in the urinary tract, respiratory tract, and bloodstream infections, particularly in patients with serious illnesses (3). Ceftazidime–avibactam (CZA) and tigecycline (TGC) are used as the last-line choices for the treatment of carbapenem-resistant *K. pnuemoniae* (CRKP) infections (2, 4). CRKP is based on resistance to any kind of carbapenems. However, acquisition of CZA resistance has been reported after CZA therapy in recent years (5). The resistance to CZA has been observed in strains with mutations in *amp*C, *bla*$_{KPC-2}$, and *bla*$_{KPC-3}$, and the mutation points were mostly in the Ω-loop in *bla*$_{KPC}$ genes (4, 6–8). In addition, the high expression of KPC-3 was reported to be associated with CZA resistance (4, 9), which could also be induced by alteration or disruption of OmpK35/36 (10, 11).

The CRKP strains of the hyperepidemic clonal complex 258 are detected worldwide as hospital-acquired pathogens and are frequently responsible for outbreaks (4, 12). In particular, the sequence type 258 (ST258), ST512, ST340, and ST11 are the most frequently detected variants of KPC-producing *K. pneumoniae* isolates (2, 4). The ST11 *K. pneumoniae* strains are endemic to East Asia, especially in China (13), while the others are pandemic in western countries (14, 15).

In this study, we present an IncFIB plasmid that carried double copies of *bla*$_{KPC-2}$ and *bla*$_{SHV-182}$ and four common resistance genes (including *bla*$_{CTX-M-65}$, *bla*$_{TEM-1}$, and *rmt*B) from a colistin- and tigecycline-resistant *K. pneumoniae* strain Kpn_XM9. The aim of this study was to describe the phenotypic and genotypic adaptive characteristics of Kpn_XM9. It suggests that we should pay more attention to the CRKP harboring two or more *bla*$_{KPC-2}$ genes, which greatly increase the minimum inhibitory concentration (MIC) of CZA.

## MATERIALS AND METHODS

### Clinical case information and isolate collection

The strain Kpn_XM9 was obtained on 22 July 2022 from the urine sample of an 89-year male patient with pneumonia, cerebral hemorrhage, and gastrointestinal bleeding. The patient was hospitalized at the intensive care unit of Xiamen Humanity Hospital Fujian Medical University. Before isolation, the patient was once treated with meropenem, linezolid, tigecycline, polymyxin B, cefoperazone, caspofungin, moxifloxacin, and imipenem, while the history of the CZA regimen was not found.

### Whole-genome sequencing and analysis

The genomic DNA of the Kpn_XM9 strain was prepared using the QIAamp DNA mini kit (Qiagen, Hilden, Germany) and was subjected to both Illumina paired-end sequencing (Illumina Inc., San Diego, CA) and long-read nanopore sequencing (Oxford Nanopore Technologies, Oxford, UK). The whole genome was assembled by Unicycler v0.5.0, and cyclization and calibration of the assembled genome were performed by Circulator v0.5.0 and Polypolish v0.5.0, respectively. The assembled genome was subsequently annotated with the Prokka pipeline.

The resistance genes were detected by ResFinder 3.2 with a 90% threshold for gene identification and a 60% minimum length coverage. The virulence genes were identified by querying the database available at http://cge.food.dtu.dk/services/VirulenceFinder/ with a 90% threshold for gene identification and a 60% minimum length coverage.

The serotype was predicted at http://bigsdb.pasteur.fr/cgi-bin/bigsdb/bigsdb.pl?db=pubmlst_klebsiella_seqdef&page=sequenceQuery. Multilocus sequence typing (MLST) was performed with the Center for Genomic Epidemiology guidelines (http://cge.cbs.dtu.dk/services/MLST/). The plasmid replicon type was determined using the PlasmidFinder tools at http://cge.food.dtu.dk/services/PlasmidFinder/. The genomic

islands, insertion sequence (IS) elements, restriction and modification (R-M) sequences, clustered regularly interspaced short palindromic repeat (CRISPR) sequences, and secondary metabolite gene clusters were predicted by IslandViewer4, ISFinder1.0, Restriction-ModificationFinder, CRISPRCasFinder1.0, and antiSMASH7.1.0, respectively.

## Mating test

The conjunction experiments were carried out in broth and on filters with the azide-resistant *E.coli* strain J53 as the recipient at both 25°C and 37°C. The potential transconjugants were selected on Luria–Bertani agar plates containing both 4 µg/mL meropenem and 150 µg/mL azide.

## Transposon deletion

Transposon deletion was performed as previously described (16). The sequences and sources of primers are shown in Table 1. Kpn_XM9 was converted into the competent state, followed by electrotransformation of Plasmid pKOBEG. After selection under 50 µg/mL apramycin, the Kpn_XM9 strain with pKOBEG was induced by arabinose and further converted into the competent state; a homogenous fragment of 3,408 bp was then electrotransformed into it to perform transposon deletion. The *catA* gene was amplified from plasmid pKD3. The TnAs1-knockout was selected under 100 µg/mL chloramphenicol. The knockouts were confirmed by sequencing the fragment a little longer than the target sites.

## Antimicrobial susceptibility testing

MICs of amikacin, aztreonam, ticarcillin/clavulanic acid, piperacillin/tazobactam, ceftazidime, cefepime, imipenem, meropenem, tobramycin, ciprofloxacin, levofloxacin, doxycycline, minocycline, tigecycline, colistin, and trimethoprim/sulfamethoxazole were determined using the broth microdilution method as per the Clinical and Laboratory Standards Institute (CLSI) (17). The MIC of CZA was determined using the E-test method as per the CLSI. As there is no breakpoint for tigecycline from CLSI, that defined by the European Committee on Antimicrobial Susceptibility Testing (EUCAST) (http://www.eucast.org/) was applied. Antimicrobial susceptibility testing was done in triplicate.

## RESULTS

### General traits of Kpn_XM9

The Kpn_XM9 belonged to K64 and ST11, had only Type II R-M (recognition sequence: CCWGG) system, but no CRISPR–Cas system. The predicted virulence genes are shown in Table 2: it is concluded that Kpn_XM9 had Types 1 and 3 fimbriae, regular capsule, efflux pump, four siderophores (enterobactin, yersiniabactin, salmochelin, and aerobactin), Type VI secretion system, and lipopolysaccharides but no allantoin utilization system, regulators of mucoid phenotype, and colibactin. The drug resistance genes were carried by chromosome, p1_Kpn_XM9, and p3_Kpn_XM9 (Table 3). In particular, $bla_{KPC-2}$ was on p1_Kpn_XM9. In addition, Table 4 showed that p1_Kpn_XM9 and p2_Kpn_XM9 were mobilizable and p3_Kpn_XM9 was conjugative, while p4_Kpn_XM9 and p5_Kpn_XM9 were both unmobilizable (18).

**TABLE 1**  The primers used in this study[a]

| Primer | Sequence (5′–3′) | Tm (°C) | Product (bp) |
|---|---|---|---|
| $bla_{KPC-2}$ up F | ACGCATTGAATTCTTCCGGT | 65 | 1,193 |
| $bla_{KPC-2}$ up R | AACTAAGGAGGATATTCATATGGCGGCGCTACATCAGGTTGCA | 71 | |
| $bla_{KPC-2}$ down F | GAAGCAGCTCCAGCCTACACTTTCTCTACGATCCAGCGCA | 66 | 1,200 |
| $bla_{KPC-2}$ down R | CGCTGCCATATCCCGTTACC | 68 | |
| *catA* F | CCATATGAATATCCTCCTTAGTT | 58 | 1,015 |
| *catA* R | GTGTAGGCTGGAGCTGCTTC | 68 | |

[a]Tm: melting temperature; bp: base pair. Tm values were calculated at http://tmcalculator.neb.com/#!/main.

**TABLE 2** Virulence genes found in Kpn_XM9[a]

| Function | Virulence factor | Location | | | | | |
|---|---|---|---|---|---|---|---|
| | | Chromosome | Plasmid 1 | Plasmid 2 | Plasmid 3 | Plasmid 4 | Plasmid 5 |
| Adherence | Type 3 fimbriae | *mrk*A-D, *mrk*F, and *mrk*H-J | None | None | None | None | None |
| | Type 1 fimbriae | *fim*A-I and *fim*K | None | None | None | None | None |
| | Type IV pili biosynthesis | None | None | None | *pil*U | None | None |
| Antiphagocytosis | Capsule | *wzi* | None | None | None | None | None |
| Efflux pump | AcrAB | *acr*A-B | None | None | None | None | None |
| Iron uptake | Aerobactin | *iut*A | None | *iuc*A-D and *iut*A | None | None | None |
| | Enterobactin | *ent*A-F, *ent*S, *fep*A-D, *fep*G, and *fes* | None | None | None | None | None |
| | Salmochelin | *iro*E and *iro*N | None | None | None | None | None |
| | Yersiniabactin | *fyu*A, *irp*1-2, *ybt*A, *ybt*E, *ybt*P-Q, *ybt*S-U, and *ybt*X | None | None | None | None | None |
| Nutritional factor | Allantoin utilization | None | None | None | None | None | None |
| Regulation | RcsAB | *rcs*A-B | None | None | None | None | None |
| | RmpA | None | None | None | None | None | None |
| Secretion system | T6SS-I | *clp*V/*tss*H, *dot*U/*tss*L, *hcp*/*tss*D, *icm*F/*tss*M, *imp*A/*tss*A, *omp*A, *sci*N/*tss*J, *tli*1, *tss*F-G, *vas*E/*tss*K, *vgr*G/*tss*I, and *vip*B/*tss*C | None | None | None | None | None |
| | T6SS-II | *clp*V | None | None | None | None | None |
| | T6SS-III | *icm*F, *imp*A, *imp*F, *imp*G, *imp*H, and *sci*N | None | None | None | None | None |
| Serum resistance | LPS | *rfb* | None | None | None | None | None |
| Toxin | Colibactin | None | None | None | None | None | None |

[a]T6SS: Type Ⅵ secretion system; LPS: lipopolysaccharide.

## Genomic structural analysis

The chromosome mapping of Kpn_XM9 is shown in Fig. 1 with HS11286 being the control and drug-resistance and virulence genes being presented. HS11286 (GenBank accession No. CP003200.1) was one *K. pneumoniae* strain isolated from sputum and presented serotype 47, ST11, and carbapenem resistance, which is due to $bla_{KPC-2}$. The chromosome of Kpn_XM9 was 5,523,536 bp, which harbored drug resistance genes *aad*A2b, $bla_{SHV-182}$, *sul*1, and *fosA*6 and virulence genes *mrk*A-D, *mrk*F, *mrk*H-J, *fim*A-I, *fim*K, *wzi, iut*A, *ent*A-F, *ent*S, *fep*A-D, *fep*G, *fes, iro*E, *iro*N, *fyu*A, *irp*1-2, *ybt*A, *ybt*E, *ybt*P-Q, *ybt*S-U, *ybt*X, *rcs*A-B, and *rfb*.

The mapping of p1_Kpn_XM9 is shown in Fig. 2, with p3_L39 (GenBank ID CP033956.1) and p17ZR-91-TC1 (MN200130.1) being compared. The plasmid p1_Kpn_XM9 was 128,219 bp in size and belonged to the IncFIB type. It harbored drug resistance genes *rmt*B, $bla_{KPC-2}$, $bla_{CTX-M-65}$, and $bla_{TEM-1B}$ but no virulence genes. It has 12 IS26, six TnAs1, and two copies of $bla_{KPC-2}$ and $bla_{SHV-182}$.

**TABLE 3** Drug resistance genes harbored by Kpn_XM9

| Location | Drug resistance genes |
|---|---|
| Chromosome | *aad*A2b, $bla_{SHV-182}$, *sul*1, and *fosA*6 |
| Plasmid 1 | *rmt*B, $bla_{KPC-2}$, $bla_{CTX-M-65}$, and $bla_{TEM-1B}$ |
| Plasmid 2 | None |
| Plasmid 3 | *qnr*S1, $bla_{LAP-2}$, *sul*2, *dfr*A14, and *tet*A |
| Plasmid 4 | None |
| Plasmid 5 | None |

**TABLE 4** Prediction of mobilities of five plasmids in Kpn_XM9[b]

| Location | oriT | Relaxase | T4CP | T4SS | Mobility |
|---|---|---|---|---|---|
| Plasmid 1 | 18575–18660[a] | None | None | 115279–121580[a] | Mobilizable |
| Plasmid 2 | 40349–40376[a] | 93114–93845[a] | 91150–93114[a] | None | Mobilizable |
| Plasmid 3 | 5724–5805[a] | 6161–8089[a] | 36849–39041[a] | 13948–39041[a] | Conjugative |
| Plasmid 4 | None | None | None | None | Unmobilizable |
| Plasmid 5 | None | None | None | None | Unmobilizable |

[a]Represents the exact locations on the five plasmids.
[b]T4CP: Type IV coupling protein; T4SS: Type IV secretion system.

The resistance regions of p1_Kpn_XM9 are shown in Fig. 3, with p3_L39 and p17ZR_91_TC1 being compared. Gene $bla_{KPC-2}$ is located in TnAs1, flanked by two copies of IS26 mobile elements (IS26-ISKpn27-$bla_{KPC-2}$-ISKpn6-TnAs1-$bla_{SHV-182}$-IS26).

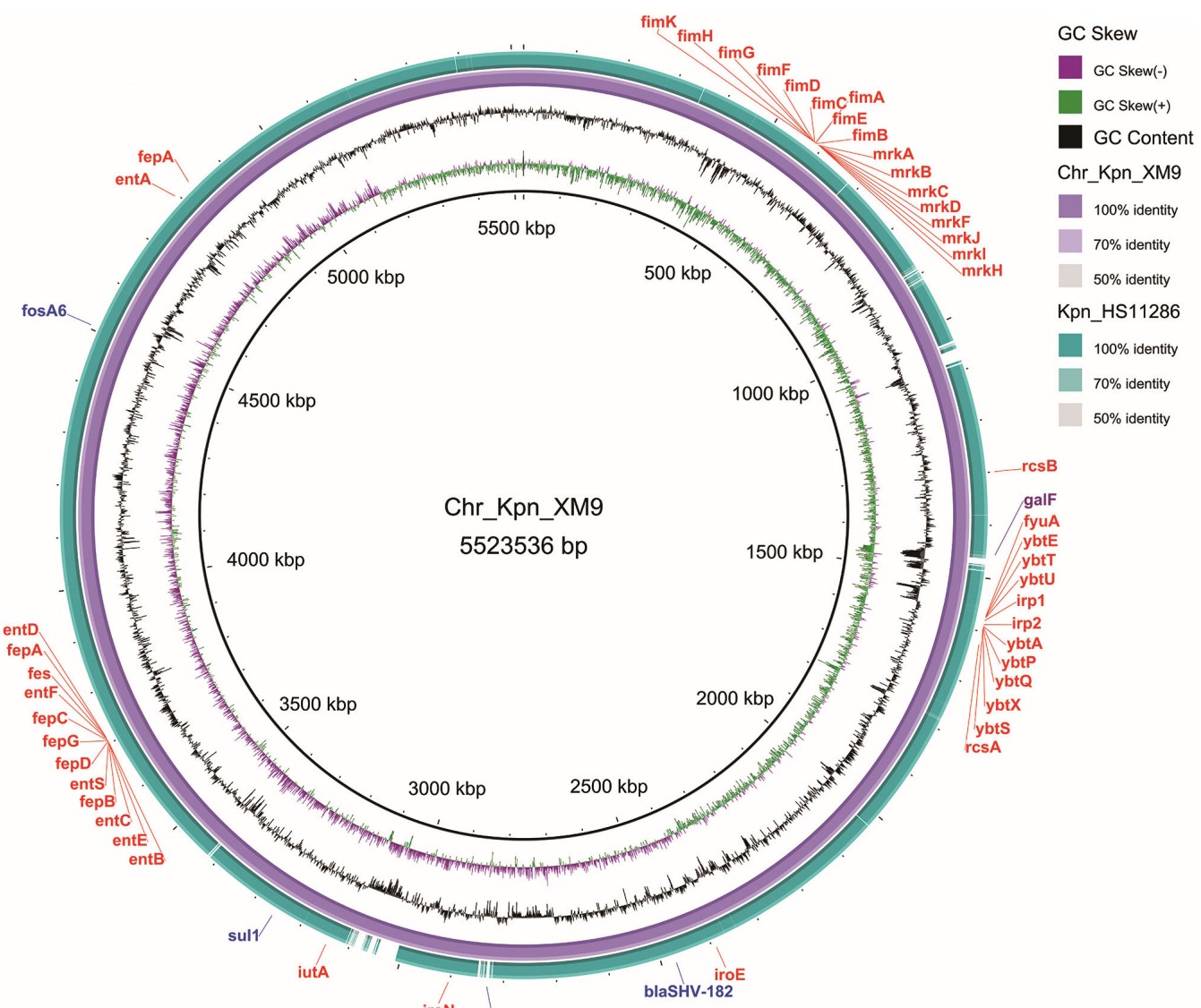

**FIG 1** Circular map for genetic characterization of the Kpn_XM9 chromosome. Sequence comparison of Chr_Kpn_XM9 (GenBank accession number: CP140645) with other completely sequenced chromosome: HS11286 (reference genome ASM24018v2). The map was drawn using BRIG. The resistance and virulence genes have been labeled. The blue, red, and purple squares show the resistance genes, virulence genes, and capsule gene of the Chr_Kpn_XM9 chromosome, respectively.

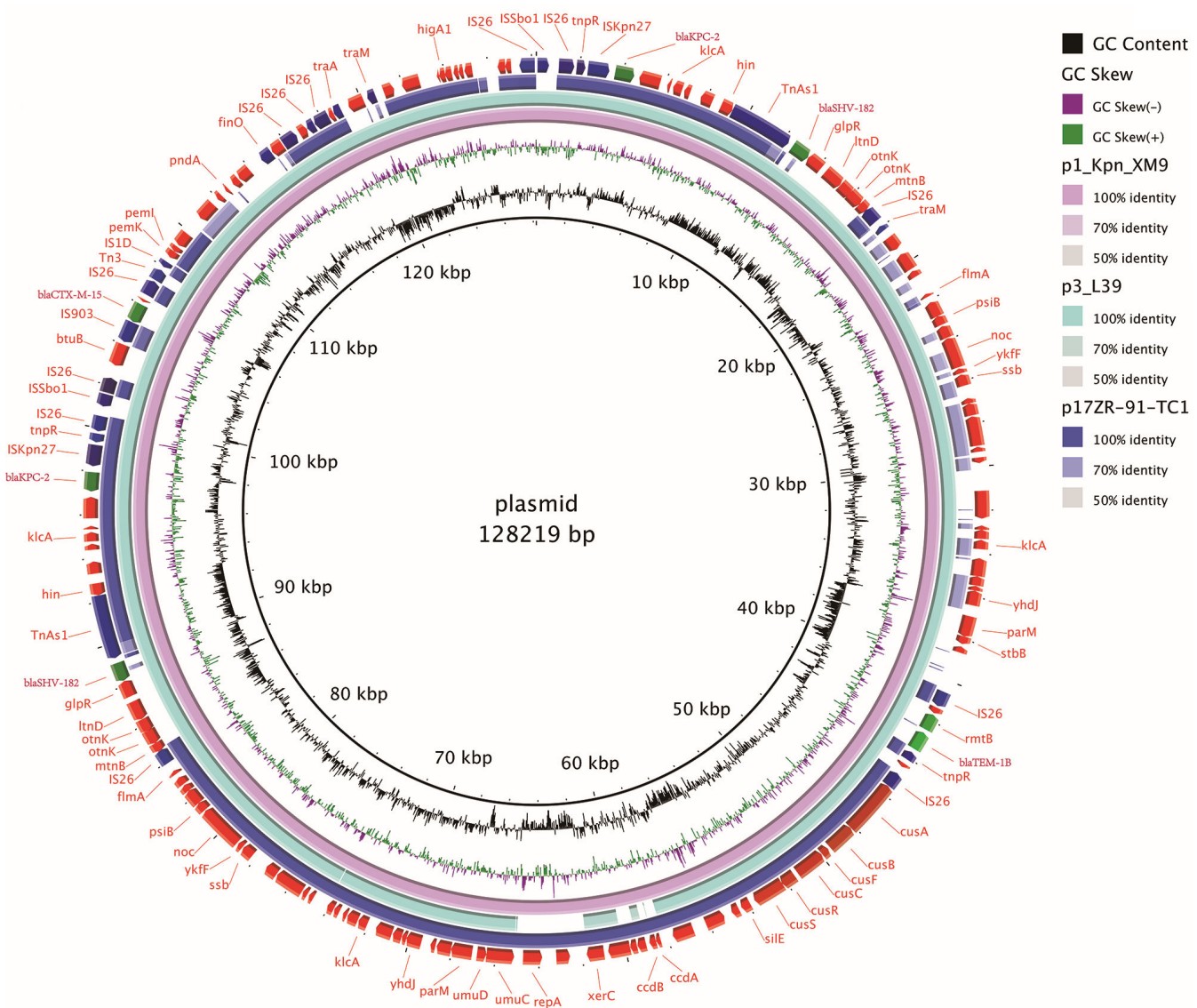

**FIG 2** Genetic characterization of the IncFIB plasmid p1_Kpn_XM9. Circular map of the plasmid p1_Kpn_XM9 in comparison with similar reported plasmids using BRIG. The analyzed plasmids were p3_L39 (GenBank accession number: CP033956.1) and p17ZR-91-TC1 (GenBank accession number: MN200130.1). The blue, green, and purple squares show the mobile elements, resistance genes, and type IV secretion system of the p1_Kpn_XM9 plasmid, respectively.

Two copies of $bla_{KPC-2}$ are present at a distance of 92,566 bp within the IncFIB plasmid scaffold of p1_Kpn_XM9.

## Antimicrobial susceptibility testing results

As shown in Table 5, Kpn_XM9 was resistant to all the tested antimicrobials, except CZA. Noticeably, the Kpn_XM9 retained the aforementioned resistance: the MIC of CZA declined from 8/4 to 2/4 µg/mL, when the 18,555 bp nucleotide harboring $bla_{KPC-2}$ was knocked out.

After several conjugation tests, no conjugants were found resistant to meropenem and azide, which confirmed the unconjugative property of p1_Kpn_XM9.

## DISCUSSION

The antimicrobial resistance spectrum of Kpn_XM9 confirmed that it was an extreme drug-resistant bug (19), provided its susceptibility to CZA and chloramphenicol (data

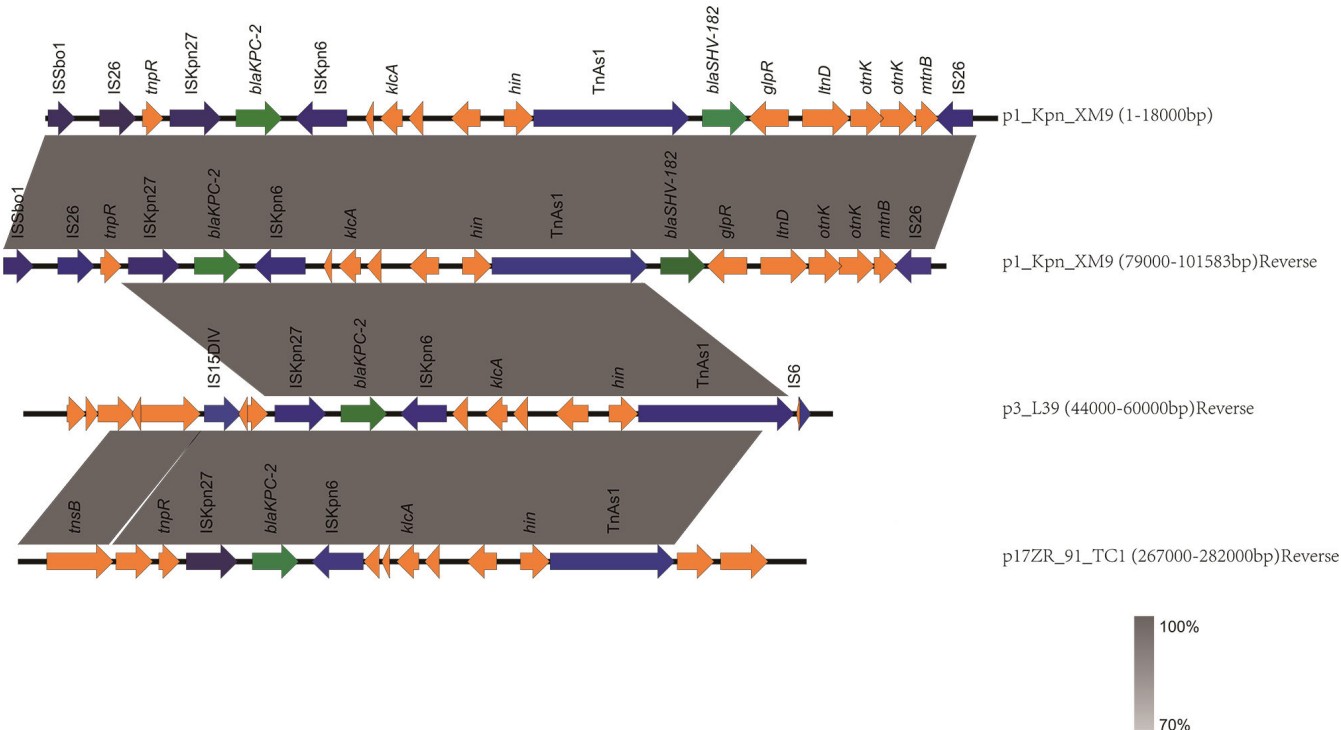

**FIG 3** Comparison of the resistant regions of plasmids p1_Kpn_XM9, p3_L39, and p17ZR_91_TC1. Gray shading denotes shared regions of homology (> 99.9% nucleotide similarity). Open-reading frames are indicated by arrows and colored based on predicted gene function. Antibiotic resistance genes and mobile elements are labeled in green and blue, respectively. Other genes in the accessory region are indicated by yellow arrows.

not shown). The Kpn_XM9 strain raised a great clinical challenge for the difficult choice of antibiotics. Chloramphenicol is nowadays an uncommon clinical choice due to its toxicities, e.g., bone marrow toxicity (20). As strains of CRKP become more prevalent around the world, clinical choices become more and more limited, e.g., colistin, tigecycline, and CZA (21). The CZA is used as the most favored choice, and the acquired resistance to it commonly results from mutations of $bla_{KPC-2}$ and $bla_{KPC-3}$, e.g., $bla_{KPC-33}$ and $bla_{KPC-90}$ (4). The deletion of one $bla_{KPC-2}$ conferred increased susceptibility of Kpn_XM9 to CZA, with the MIC shifting from 8/4 to 2/4 µg/mL (Table 5) as the simultaneous deleted one $bla_{SHV-182}$ played no role in susceptibility to CZA (22, 23); SHV-182 is an SHV-11 variant with an S106T mutation, and SHV-11 makes no contribution to resistance to CAZ or avibactam. To date, more than two copies of $bla_{KPC-2}$ are much rare in *K. pneumoniae* (24); Strain WCHKP2 was ST11 and K64 and had three copies of $bla_{KPC-2}$ on a plasmid with an IncR and an IncFII replicon. In addition, the plasmid was also not self-transmissible. Apart from drug resistance, although Kpn_XM9 harbored various virulence genes, it could not yield a hypercapsule due to the absence of related regulators and therefore be denoted as classical *K. pneumoniae* (25, 26).

Genomic analysis revealed a genome with 57.3% GC content and the presence of an IncFIB plasmid (p1_Kpn_XM9) with 53.8% GC content and an IncHI1B plasmid (p2_Kpn_XM9) with 50.4% GC content. Strain Kpn_XM9 is the predominant ST of CRKP in

**TABLE 5** Antimicrobial susceptibility testing results of Kpn_XM9 and its knockout[a]

| Strain | TIC | TZP | CAZ | SCF | FEP | ATM | IMI | MEM | AK | TOB | CIP | LEV | DOX | MN | TGC | COL | SXT | CZA |
|---|---|---|---|---|---|---|---|---|---|---|---|---|---|---|---|---|---|---|
| XM9 | ≥128/2 | ≥128/4 | ≥64 | ≥64/32 | ≥64 | ≥64 | ≥16 | ≥16 | ≥64 | ≥16 | ≥4 | ≥8 | ≥16 | ≥16 | ≥8 | 8 | ≥16/304 | 8/4 |
| Knockout | ≥128/2 | ≥128/4 | ≥64 | ≥64/32 | ≥64 | ≥64 | ≥16 | ≥16 | ≥64 | ≥16 | ≥4 | ≥8 | ≥16 | ≥16 | ≥8 | 8 | ≥16/304 | 2/4 |

[a]TIC: ticarcillin/clavulanic acid; TZP: piperacillin/tazobactam; CAZ: ceftazidime; SCF: cefoperazone/sulbactam; FEP: cefepime; ATM: aztreonam; IMI: imipenem; MEM: meropenem; AK: amikacin; TOB: tobramycin; CIP: ciprofloxacin; LEV: levofloxacin; DOX: doxycycline; MN: minocycline; TGC: tigecycline; COL: colistin; SXT: trimethoprim/sulfamethoxazole; CZA: ceftazidime/avibactam.

China (4, 27, 28). With respect to virulence, Kpn_XM9 had genes encoding yersiniabactin on the chromosome, enterobactin on the chromosome, aerobactin on the chromosome and p2_Kpn_XM9, and outer membrane protein on the chromosome. A total of 5,797 protein-coding sequences, 86 tRNA genes, 25 rRNA genes, and a tmRNA gene were identified. The genome also contained multiple IS elements, the majority of which belonged to the IS1, IS3, IS5, and IS6 families. No CRISPR sequence and only the Type II R-M system were predicted, which suggested its immune deficiency, and is in line with the report (29).

The Kpn_XM9 strain had seven types of antimicrobial resistance genes mediating resistance to aminoglycosides, β-lactams, fosfomycin, trimethoprim, tetracycline, ciprofloxacin, and sulfonamides. Among these resistance genes, $aad$A2b, $bla_{SHV-182}$, $sul$1, and $fosA$6 were located on the chromosome (Table 2; Fig. 1). The resistance genes $rmt$B, $bla_{KPC-2}$, $bla_{CTX-M-65}$, and $bla_{TEM-1B}$ were carried by p1_Kpn_XM9 (Table 2; Fig. 2), while the remaining genes were located on p3_Kpn_XM9 (Table 2).

As shown in Fig. 2, our results indicated that p1_Kpn_XM9 shows query coverage (95%) and nucleotide identity (100%) with the $bla_{TEM-1}$, $bla_{KPC-2}$, and $bla_{SHV-12}$ encoding plasmid p3_L39 (GenBank accession no. CP033956.1) of the *K. pneumoniae* strain isolated from stool samples in China. In addition to the plasmid p3_L39, the plasmid p1_Kpn_XM9 also had similar gene sequences with the plasmid p17ZR-91-TC1 (GenBank accession no. MN200130.1).

IS26 is the most common vector to confer $bla_{KPC-2}$ replication transposition (24, 30, 31). We observed that two copies of $bla_{KPC-2}$ were present at a distance of 92,566 bp within the IncFIB plasmid scaffold of p1_Kpn_XM9 (Fig. 3). Noticeably, there are two copies of $bla_{KPC-2}$ and $bla_{SHV-182}$ on p1_Kpn_XM9. There is a tandem repeat sequence in p1_Kpn_XM9, which is opposite to each other and contains the $bla_{KPC-2}$ and $bla_{SHV-182}$ genes. The $bla_{KPC-2}$ gene is bracketed by the ISKpn7 (upstream) and the ISKpn6 (downstream) within a Tn4401a transposon, as first described by Iida S *et al*. (32). Fortini reported the presence of double copies of $bla_{KPC-3}$ in the ST512 *K. pneumoniae*; they observed that two copies of $bla_{KPC-3}$ were present at a distance of 8,658 bp within the IncX3 plasmid scaffold (7, 9), while there was a long distance of 93,447 bp between the two $bla_{KPC-2}$ genes in this study.

The insertion of IS26 could generate 14 bp direct target repeats (DR), which were present around every copy of IS26. This suggests that the tandem multiplication of IS26-ISKpn27-$bla_{KPC-2}$-ISKpn6 was due to the composite transposon of two copies of IS26, which may be excised from plasmids to form a circular intermediate (7, 33). Sometimes there is no DR element near the IS26 transposon, in which case the diffusion of the tandem element (IS26-ISKpn27-$bla_{KPC-2}$-ISKpn6) in the bacteria can be carried out by homologous recombination. IS26 provides a region for homologous recombination and therefore could serve as a Trojan horse (31, 34). In the presence of IS26 on a plasmid, the intermediate may be integrated into the plasmid via homologous recombination to generate tandem repeats (35, 36).

Despite constant attempts, no transconjugants resistant to CZA were selected, showing that p1_Kpn_XM9 was not self-transmissible or the conjugation rate is too slight, which is in line with the prediction (Table 3), although two conjugative genes were present on p1_Kpn_XM9, $tra$A, and $tra$M, which are indispensable for the conjugative transfer process (18). The expressions of TraA anf TraM may be influenced by the unique RNA–protein interaction of FinO (37, 38), which is called IncF plasmid conjugative transfer fertility inhibition protein. This may explain why p1_Kpn_XM9 was not self-transmissible. In addition, p1_Kpn_XM9 lacked T4SS, which is essential for self-mobility (18). On the other hand, although p1_Kpn_XM9 was not self-transmissible, it was mobilizable (Table 4) and Kpn_XM9 had a conjugative IncFII plasmid, which may act as a helper of its mobilization (39, 40) and should be of great concern.

## Conclusions

In conclusion, a novel IncFIB plasmid mediating higher MIC of CZA was described in this study, which coharbored $bla_{TEM-1B}$, $rmtB$, double copies of $bla_{KPC-2}$, and $bla_{SHV-182}$. It likely originated by recombination with elements frequently associated with IncF plasmids. More than two copies of $bla_{KPC-2}$ pose a novel therapeutic challenge and should be of great concern.

## ACKNOWLEDGMENTS

This work was supported by the Zhejiang Provincial Health Commission (grant numbers 2023KY1326 and 2024KY535). The funding source had no role in the study design; collection, analysis, and interpretation of data; the writing of the report; and the decision to submit the article for publication.

D.H., S.W., and M.X. conceived the study. S.W. and X.M. collected and identified the strain. D.H., J.Z., and X.L. performed gene deletions. J.Z., X.L., and Q.M. performed antimicrobial susceptibility tests. D.H., X.L., and W.Z. performed bioinformatic analysis. D.H., S.W., and M.X. drafted the manuscript, which was revised by Q.M. and X.M.

## AUTHOR AFFILIATIONS

[1]Department of Laboratory Medicine, Taizhou Municipal Hospital, Taizhou, Zhejiang, China
[2]Department of Clinical Laboratory, Xiamen Humanity Hospital Fujian Medical University, Xiamen, Fujian, China
[3]Department of Pathology, Zhongshan Hospital, Fudan University, Shanghai, China
[4]Xiamen Key Laboratory of Genetic Testing, Department of Clinical Laboratory, the First Affiliated Hospital of Xiamen University, Xiamen, Fujian, China

## AUTHOR ORCIDs

Qinfei Ma http://orcid.org/0009-0005-4509-6928
Xiaobo Ma http://orcid.org/0000-0002-2150-5062

## FUNDING

| Funder | Grant(s) | Author(s) |
|---|---|---|
| Zhejiang Provincial Health Commission | 2023KY1326 | Dakang Hu |
| Zhejiang Provincial Health Commission | 2024KY535 | Dakang Hu |

## DATA AVAILABILITY

The complete sequences of p1_Kpn_XM9, p2_Kpn_XM9, p3_Kpn_XM9, p4_Kpn-XM9, p5_Kpn_XM9, and the chromosome of strain Kpn_XM9 have been deposited into GenBank under the accession no. CP140645, CP140650, CP140646, CP140647, CP140648, and CP140649.

## ETHICS APPROVAL

The ethical approval was obtained at Xiamen Humanity Hospital Fujian Medical University (approval number: HAXM-MEC-20240112-004-01).

## ADDITIONAL FILES

The following material is available online.

## Open Peer Review

**PEER REVIEW HISTORY (review-history.pdf).** An accounting of the reviewer comments and feedback.

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
