## [Reviewer comments · Microbiology Spectrum]

Microbiology Spectrum

Double *bla*_{KPC-2} copies quadrupled minimum inhibitory concentration of ceftazidime-avibactam in hospital-derived *Klebsiella pneumoniae*

Dakang Hu, Shijie Wang, Mengqiao Xu, Jin Zhang, xinhua luo, Wei Zhou, Qinfei Ma, and Xiaobo Ma

Corresponding Author(s): Xiaobo Ma, 1st Affiliated Hospital of Xiamen University

Review Timeline:

Submission Date:	February 5, 2024
Editorial Decision:	March 25, 2024
Revision Received:	April 11, 2024
Editorial Decision:	May 14, 2024
Revision Received:	May 17, 2024
Accepted:	May 30, 2024

Editor: Brian Conlon

Reviewer(s): The reviewers have opted to remain anonymous.

Transaction Report:

DOI: <https://doi.org/10.1128/spectrum.00331-24>

Re: Spectrum00331-24 (Double *bla*_{KPC-2} copies quadrupled minimum inhibitory concentration of ceftazidime-avibactam in hospital-derived *Klebsiella pneumoniae*)

Dear Mr. Xiaobo Ma:

Thank you for the privilege of reviewing your work. Below you will find my comments, instructions from the Spectrum editorial office, and the reviewer comments.

The reviewers both recommend changes, in particular the reviewers noted a lack of methodological information. Please address all the reviewers concerns before resubmitting a revised manuscript.

Revision Guidelines

Sincerely,
Brian Conlon
Editor
Microbiology Spectrum

Reviewer #1 (Comments for the Author):

Hu and colleagues present an interesting report of a clinical isolate of classical carbapenem-resistant *Klebsiella pneumoniae* with broad antibiotic resistance features, but most notably elevated resistance towards ceftazidime-avibactam. Because of the clinical utility of ceftazidime-avibactam for challenging to treat *K. pneumoniae* and other Enterobacterales, understanding

mechanisms of resistance is important.

Beyond descriptions of the genomic features, the authors work to demonstrate that a repeated region of a plasmid which encodes two copies of blaKPC-2 is responsible for elevated ceftazidime-avibactam MIC. This relies on data from a mutant of the p1_Kpn_XM9 plasmid lacking 20,947 bp region including blaKPC-2. The evidence for this conclusion is diminished for two reasons: 1) lack of experimental detail and 2) the deletion of the region also deletes a copy of blaSHV-182 (if I understand correctly).

Experimental detail: The methods of deletion are lacking. The only reference included in line 128 provides details for E. coli K12 using lambda recombinase. More details are needed for clarity including the antibiotic selection, source of lambda recombinase, and what exact region of the plasmid was deleted (which could be denoted in Figures 2-3). I attempted to map the primers in Table 1 to the plasmid sequence (lines 143-146) to better understand the deletion, but was unable to find the sequence in Genbank. Additionally, because the duplicated regions are highly homologous, understanding how the deletion strategy was specified to one of the two homologous regions will be helpful. Does PCR and sequencing confirm retention of the second locus containing KPC-2?

SHV-182: Like the authors, I agree that the second KPC-2 is likely responsible for an increase in CZA MIC. However, the experimental strategy to delete the second copy of SHV-182 complicates this conclusion. A more straightforward deletion of just one allele of KPC-2 while retaining both plasmid encoded SHV-182 would provide stronger support.

Additional comments:

1. Methods of transconjugation (lines 122-125) are lacking. Importantly, line 124-125 states that transconjugants were selected with meropenem which seems inappropriate to identify transconjugants with elevated CZA resistance (lines 186-187).
2. Please define CRKP in line 69
3. In Figure 2, it would clarify the interpretation if KPC-2 and SHV-182 were labelled as such instead of KPC and SHV-2 as current
4. Line 163: what is HS11286, what is its source, and why is it a comparator?
5. The MIC of the isolate is listed as 8 ug/ml for CZA. The reader assumes 4 µg/ml avibactam with 8 µg/ml ceftazidime but it is not specified. There is a similar lack of reporting the beta-lactamase inhibitor concentration in Table 5 for TIC, TZP, SCF.
6. Information on the number of replicates and reproducibility of the MIC determination would be helpful.

Reviewer #2 (Comments for the Author):

The authors analyzed KPC-producing *K. pneumoniae* strain Kpn_XM9, which showed an increased MIC to ceftazidime-avibactam (CZA). Whole genome sequencing analysis revealed that Kpn_XM9 is capsular genotype K64, sequence type (ST) 11, and harbors two copies of blaKPC-2 on a single plasmid. After knockout of one of the two copies of blaKPC-2 in the strain, the MIC of CZA was reduced from 8 ug/mL to 2 ug/mL.

Although the authors' observations are interesting, additional analysis is needed to demonstrate a causal relationship between the observed findings. In addition, a more detailed description of the methodology of the analysis needs to be added and the presentation of the results needs to be modified.

Major comments

- Since the knockout experiment of one copy of blaKPC-2 is the most important component of the present report, it is desirable to describe the experimental method in more detail. In addition, the amount of blaKPC transcription must be reduced in the knockout strain for the copy number change to affect the antimicrobial susceptibility. This should be confirmed experimentally. In the Discussion section, similarities and differences in the genetic background and effects on antimicrobial susceptibility should be discussed with previous reports of *K. pneumoniae* strains possessing multiple copies of blaKPC (PMID: 26525794, 30496820, etc.).
- One of the characteristics of the strains analyzed in this study was carriage of many virulence genes. The spread of hypervirulent KPC-producing ST11 *K. pneumoniae* strains in China has been recognized as a serious public health threat. The virulence of the analyzed strain should be explained in the Discussion section, comparing the genetic environment of the major virulence genes (*iro*, *iuc*, *ybt*) with previously detected strains.
- A lot of the text in the Discussion section (Lines 203-238) is simply a repetition of the presentation of the results and could be greatly shortened.

Minor comments

- Line 91-95: Adding a description of the background of the patient in whom the strain was detected, such as whether it was community-acquired or hospital-acquired, history of antimicrobial therapy (especially CZA), etc., will aid the readers' understanding of the factors contributing to the emergence of the strain.
- Line 136: "both microdilution method" should be corrected to "broth microdilution method".
- Line 158: The definition of "classical strain" is unclear.
- Line 158-160, Table 4: Criteria for classifying plasmid mobility are unclear.
- Line 163-164: It is unclear why the HS11286 strain was selected for comparison.
- Line 169-170: It is unclear why these two plasmids were chosen for comparison. Is there no plasmid registered so far that is an exact match to p1_Kpn_XM9, and are these two plasmids the most similar?

- Line 171-173: blaCTX-M and bla-SHV on this plasmid are described as blaToho-1 and blaSHV-2 in Fig. 2. The description should be unified to avoid confusion.
- Line 186-187: It is stated in the Materials and Methods section that the transconjugants were selected by meropenem. It is unclear from this description whether no transconjugants were obtained at all, or whether they were obtained but were susceptible to CZA.
- Line 198-201: Reference 22 reports a *Pseudomonas aeruginosa* strain that acquired CZA resistance by a mutation of one of the two copies of blaKPC-2, which is different from the resistance mechanism of the present strain.
- Line 239-241: Reference 26 is not by Nass et al.
- Line 258-259: The text appears to be truncated in the middle of the sentence.

Hu and colleagues present an interesting report of a clinical isolate of classical carbapenem-resistant *Klebsiella pneumoniae* with broad antibiotic resistance features, but most notably elevated resistance towards ceftazidime-avibactam. Because of the clinical utility of ceftazidime-avibactam for challenging to treat *K. pneumoniae* and other Enterobacterales, understanding mechanisms of resistance is important.

Beyond descriptions of the genomic features, the authors work to demonstrate that a repeated region of a plasmid which encodes two copies of blaKPC-2 is responsible for elevated ceftazidime-avibactam MIC. This relies on data from a mutant of the p1_Kpn_XM9 plasmid lacking 20,947 bp region including blaKPC-2. The evidence for this conclusion is diminished for two reasons: 1) lack of experimental detail and 2) the deletion of the region also deletes a copy of blaSHV-182 (if I understand correctly).

Experimental detail: The methods of deletion are lacking. The only reference included in line 128 provides details for *E. coli* K12 using lambda recombinase. More details are needed for clarity including the antibiotic selection, source of lambda recombinase, and what exact region of the plasmid was deleted (which could be denoted in Figures 2-3). I attempted to map the primers in Table 1 to the plasmid sequence (lines 143-146) to better understand the deletion, but was unable to find the sequence in Genbank. Additionally, because the duplicated regions are highly homologous, understanding how the deletion strategy was specified to one of the two homologous regions will be helpful. Does PCR and sequencing confirm retention of the second locus containing KPC-2?

SHV-182: Like the authors, I agree that the second KPC-2 is likely responsible for an increase in CZA MIC. However, the experimental strategy to delete the second copy of SHV-182 complicates this conclusion. A more straightforward deletion of just one allele of KPC-2 while retaining both plasmid encoded SHV-182 would provide stronger support.

Additional comments:

1. Methods of transconjugation (lines 122-125) are lacking. Importantly, line 124-125 states that transconjugants were selected with meropenem which seems inappropriate to identify transconjugants with elevated CZA resistance (lines 186-187).
2. Please define CRKP in line 69
3. In Figure 2, it would clarify the interpretation if KPC-2 and SHV-182 were labelled as such instead of KPC and SHV-2 as current
4. Line 163: what is HS11286, what is its source, and why is it a comparator?
5. The MIC of the isolate is listed as 8 ug/ml for CZA. The reader assumes 4 µg/ml avibactam with 8 µg/ml ceftazidime but it is not specified. There is a similar lack of reporting the beta-lactamase inhibitor concentration in Table 5 for TIC, TZP, SCF.
6. Information on the number of replicates and reproducibility of the MIC determination would be helpful.

Replies to reviewers' comments

Reviewer #1 (Comments for the Author):

Hu and colleagues present an interesting report of a clinical isolate of classical carbapenem-resistant *Klebsiella pneumoniae* with broad antibiotic resistance features, but most notably elevated resistance towards ceftazidime-avibactam. Because of the clinical utility of ceftazidime-avibactam for challenging to treat *K. pneumoniae* and other Enterobacterales, understanding mechanisms of resistance is important.

Beyond descriptions of the genomic features, the authors work to demonstrate that a repeated region of a plasmid which encodes two copies of *bla*_{KPC-2} is responsible for elevated ceftazidime-avibactam MIC. This relies on data from a mutant of the p1_Kpn_XM9 plasmid lacking 20,947 bp region including *bla*_{KPC-2}. The evidence for this conclusion is diminished for two reasons: 1) lack of experimental detail and 2) the deletion of the region also deletes a copy of *bla*_{SHV-182} (if I understand correctly).

Experimental detail: The methods of deletion are lacking. The only reference included in line 128 provides details for *E. coli* K12 using lambda recombinase. More details are needed for clarity including the antibiotic selection, source of lambda recombinase, and what exact region of the plasmid was deleted (which could be denoted in Figures 2-3). I attempted to map the primers in Table 1 to the plasmid sequence (lines 143-146) to better understand the deletion, but was unable to find the sequence in Genbank. Additionally, because the duplicated regions are highly homologous, understanding how the deletion strategy was specified to one of the two homologous regions will be helpful. Does PCR and sequencing confirm retention of the second locus containing KPC-2?

Reply: The gene deletion details are now added. The sequence of p1_Kpn_XM9 has now been deposited in GenBank and another public database (<https://pan.baidu.com/s/1ShkN0GBHpkCnRxKUZEJ04Q?pwd=g79y; key: g79y>). The deleted fragment was 18555 bp. Although highly homologous regions are found on p1_Kpn_XM9, the homologous fragments adjacent to them are different. PCR and sequencing were performed to confirm retention of the second locus containing *bla*_{KPC-2}.

SHV-182: Like the authors, I agree that the second KPC-2 is likely responsible for an increase in CZA MIC. However, the experimental strategy to delete the second copy of SHV-182 complicates this conclusion. A more straightforward deletion of just one allele of KPC-2 while retaining both plasmids encoded SHV-182 would provide stronger support.

Reply: Since the 2 regions are identical and the 2 *bla*_{KPC-2} are among them, it is impractical to carry out such a deletion regardless of λ -red or CRISPR-Cas9 methods. In addition, SHV-182 does not induce resistance to carbapenem or CZA.

Additional comments:

1. Methods of transconjugation (lines 122-125) are lacking. Importantly, line 124-125 states that transconjugants were selected with meropenem which seems inappropriate to identify transconjugants with elevated CZA resistance (lines 186-187).

Reply: The potential transconjugants were selected on Luria-Bertani agar plates containing **both** 4 µg/ml meropenem and 150 µg/ml azide.

2. Please define CRKP in line 69.

Reply: The definition is now added.

3. In Figure 2, it would clarify the interpretation if KPC-2 and SHV-182 were labelled as such instead of KPC and SHV-2 as current

Reply: Genes *bla*_{KPC-2} and *bla*_{SHV-182} are now added.

4. Line 163: what is HS11286, what is its source, and why is it a comparator?

Reply: The introduction of HS11286 is now added.

5. The MIC of the isolate is listed as 8 µg/ml for CZA. The reader assumes 4 µg/ml avibactam with 8 µg/ml ceftazidime but it is not specified. There is a similar lack of reporting the beta-lactamase inhibitor concentration in Table 5 for TIC, TZP, SCF.

Reply: The MICs of the isolate for CZA, TIC, TZP, SCF, and SXT are now updated.

6. Information on the number of replicates and reproducibility of the MIC determination would be helpful.

Reply: It is now stated: Antimicrobial susceptibility testing was done in thrice.

Reviewer #2 (Comments for the Author):

The authors analyzed KPC-producing *K. pneumoniae* strain Kpn_XM9, which showed an increased MIC to ceftazidime-avibactam (CZA). Whole genome sequencing analysis revealed that Kpn_XM9 is capsular genotype K64, sequence type (ST) 11, and harbors two copies of *bla*_{KPC-2} on a single plasmid. After knockout of one of the two copies of *bla*_{KPC-2} in the strain, the MIC of CZA was reduced from 8 µg/mL to 2 µg/mL.

Although the authors' observations are interesting, additional analysis is needed to demonstrate a causal relationship between the observed findings. In addition, a more detailed description of the methodology of the analysis needs to be added and the presentation of the results needs to be modified.

Major comments

- Since the knockout experiment of one copy of *bla*_{KPC-2} is the most important component of the present report, it is desirable to describe the experimental method in more detail. In addition, the amount of *bla*_{KPC} transcription must be reduced in the knockout strain for the copy number change to affect the antimicrobial susceptibility. This should be confirmed experimentally. In the Discussion section, similarities and differences in the genetic background and effects on antimicrobial susceptibility should be discussed with previous reports of *K. pneumoniae* strains possessing multiple copies of *bla*_{KPC} (PMID: 26525794, 30496820, etc.).

Reply: The gene deletion details are now added. In the Discussion section, similarities

and differences in the genetic background were discussed with previous reports of *K. pneumoniae* strains possessing multiple copies of *bla*_{KPC}. However, MICs of CZA couldn't be compared for the reasons such as different conditions or the absence.

- One of the characteristics of the strains analyzed in this study was carriage of many virulence genes. The spread of hypervirulent KPC-producing ST11 *K. pneumoniae* strains in China has been recognized as a serious public health threat. The virulence of the analyzed strain should be explained in the Discussion section, comparing the genetic environment of the major virulence genes (*iro*, *iuc*, *ybt*) with previously detected strains.

Reply: In fact, only excessive siderophores do not induce hypervirulence in ST11 strain (PMID: 37065211). Therefore, Kpn_XM9 is cKP rather not HvKP.

- A lot of the text in the Discussion section (Lines 203-238) is simply a repetition of the presentation of the results and could be greatly shortened.

Reply: Lines 203-238 are now refined.

Minor comments

- Line 91-95: Adding a description of the background of the patient in whom the strain was detected, such as whether it was community-acquired or hospital-acquired, history of antimicrobial therapy (especially CZA), etc., will aid the readers' understanding of the factors contributing to the emergence of the strain.

Reply: The history of antimicrobial use is now added.

- Line 136: "both microdilution method" should be corrected to "broth microdilution method".

Reply: It is updated.

- Line 158: The definition of "classical strain" is unclear.

Reply: Two references are now cited.

- Line 158-160, Table 4: Criteria for classifying plasmid mobility are unclear.

Reply: One reference is now cited.

- Line 163-164: It is unclear why the HS11286 strain was selected for comparison.

Reply: The introduction of HS11286 is now added.

- Line 169-170: It is unclear why these two plasmids were chosen for comparison. Is there no plasmid registered so far that is an exact match to p1_Kpn_XM9, and are these two plasmids the most similar?

Reply: The reason these two plasmids are being compared is that there is no exact match for p1_Kpn_XM9 so far, and these two plasmids are most similar to the plasmid p1_KPN_XM9.

- Line 171-173: blaCTX-M and bla-SHV on this plasmid are described as blaToho-1 and blaSHV-2 in Fig. 2. The description should be unified to avoid confusion.

Reply: The descriptions are now unified in this plasmid.

- Line 186-187: It is stated in the Materials and Methods section that the transconjugants were selected by meropenem. It is unclear from this description whether no transconjugants were obtained at all, or whether they were obtained but were susceptible to CZA.

Reply: The potential transconjugants were selected on Luria-Bertani agar plates

containing **both** 4 µg/ml meropenem and 150 µg/ml azide. Line 186-187 is updated as: After several conjugation tests, no conjugants were found resistant to meropenem and azide, which confirmed the unconjugative property of p1_Kpn_XM9.

- Line 198-201: Reference 22 reports a Pseudomonas aeruginosa strain that acquired CZA resistance by a mutation of one of the two copies of blaKPC-2, which is different from the resistance mechanism of the present strain.

Reply: The sentence is deleted.

- Line 239-241: Reference 26 is not by Nass et al.

Reply: The author is updated.

- Line 258-259: The text appears to be truncated in the middle of the sentence.

Reply: The sentence is now modified.

Re: Spectrum00331-24R1 (Double *bla*_{KPC-2} copies quadrupled minimum inhibitory concentration of ceftazidime-avibactam in hospital-derived *Klebsiella pneumoniae*)

Dear Mr. Xiaobo Ma:

Thank you for the privilege of reviewing your work. Below you will find my comments, instructions from the Spectrum editorial office, and the reviewer comments.

Revision Guidelines

Sincerely,
Brian Conlon
Editor
Microbiology Spectrum

Reviewer #1 (Comments for the Author):

SHV-182: Like the authors, I agree that the second KPC-2 is likely responsible for an increase in CZA MIC. However, the experimental strategy to delete the second copy of SHV-182 complicates this conclusion. A more straightforward deletion of just one allele of KPC-2 while retaining both plasmids encoded SHV-182 would provide stronger support.

Reply: Since the 2 regions are identical and the 2 *bla*_{KPC-2} are among them, it is impractical to carry out such a deletion

regardless of λ -red or CRISPR-Cas9 methods. In addition, SHV-182 does not induce resistance to carbapenem or CZA.
Reviewer response: If SHV-182 retention is impractical, it would be good to mention that SHV-182 was also deleted in lines 197-198 and 211 and add discussion (and reference if possible) that SHV-182 will make no contribution to CZA MIC.

Line 197: length of deletion is 18555 but remains as 20947 in line 40.

Fig. 2 Genetic characterization of the IncFIB plasmid p1_Kpn_XM9 still shows SHV-2 and KPC instead of SHV-182 and KPC-2.

Reviewer #2 (Comments for the Author):

The authors revised the manuscript for acceptable quality according to the reviewers' suggestions. Only a few minor suggestions will be mentioned.

- Line 219-222: I understood the consideration on the virulence of this strain as stated in the authors' response to the reviewers' comments. I suggest that the explanation be added to the Discussion section of the manuscript along with the references.
- Line 168-169: The authors' response to the reviewer's comments states that two references regarding the definition of "classical strain" have been added, but I cannot find them. As is generally the case, the response should clearly state in which line of the revised manuscript the revision is to be found.
- Line 262-263: Although the authors' response to the reviewer's comments states that this sentence has been modified, I cannot find any change in the revised manuscript.

SHV-182: Like the authors, I agree that the second KPC-2 is likely responsible for an increase in CZA MIC. However, the experimental strategy to delete the second copy of SHV-182 complicates this conclusion. A more straightforward deletion of just one allele of KPC-2 while retaining both plasmids encoded SHV-182 would provide stronger support.

Reply: Since the 2 regions are identical and the 2 *bla*KPC-2 are among them, it is impractical to carry out such a deletion regardless of λ -red or CRISPR-Cas9 methods. In addition, SHV-182 does not induce resistance to carbapenem or CZA.

Reviewer response: If SHV-182 retention is impractical, it would be good to mention that SHV-182 was also deleted in lines 197-198 and 211 and add discussion (and reference if possible) that SHV-182 will make no contribution to CZA MIC.

Line 197: length of deletion is 18555 but remains as 20947 in line 40.

Fig. 2 Genetic characterization of the IncFIB plasmid p1_Kpn_XM9 still shows SHV-2 and KPC instead of SHV-182 and KPC-2.

All other comments of mine have been resolved.

Reply to reviewers: Round 2

Reviewer #1 (Comments for the Author):

SHV-182: Like the authors, I agree that the second KPC-2 is likely responsible for an increase in CZA MIC. However, the experimental strategy to delete the second copy of SHV-182 complicates this conclusion. A more straightforward deletion of just one allele of KPC-2 while retaining both plasmids encoded SHV-182 would provide stronger support.

Reply: Since the 2 regions are identical and the 2 *bla*_{KPC-2} are among them, it is impractical to carry out such a deletion regardless of λ -red or CRISPR-Cas9 methods. In addition, SHV-182 does not induce resistance to carbapenem or CZA.

Reviewer response: If SHV-182 retention is impractical, it would be good to mention that SHV-182 was also deleted in lines 197-198 and 211 and add discussion (and reference if possible) that SHV-182 will make no contribution to CZA MIC.

Reply: Such words and references are now added. (Line 207-210: Marked-Up Manuscript-2)

Line 197: length of deletion is 18555 but remains as 20947 in line 40.

Reply: Line 40 is now corrected. (Line 39: Marked-Up Manuscript-2)

Fig. 2 Genetic characterization of the IncFIB plasmid p1_Kpn_XM9 still shows SHV-2 and KPC instead of SHV-182 and KPC-2.

Reply: Figure 2 is now updated.

Reviewer #2 (Comments for the Author):

The authors revised the manuscript for acceptable quality according to the reviewers' suggestions. Only a few minor suggestions will be mentioned.

- Line 219-222: I understood the consideration on the virulence of this strain as stated in the authors' response to the reviewers' comments. I suggest that the explanation be added to the Discussion section of the manuscript along with the references.

Reply: Such discussion is now added in the Discussion section. (Line 213-215: Marked-Up Manuscript-2)

- Line 168-169: The authors' response to the reviewer's comments states that two references regarding the definition of "classical strain" have been added, but I cannot find them. As is generally the case, the response should clearly state in which line of the revised manuscript the revision is to be found.

Reply: Reference 27 is now added. (Line 215: Marked-Up Manuscript-2)

- Line 262-263: Although the authors' response to the reviewer's comments states that this sentence has been modified, I cannot find any change in the revised manuscript.

Reply: The sentence was "On the other hand, although not self-transmissible, p1_Kpn_XM9 was mobilizable (Table 4) and Kpn_XM9 had a conjugative IncFII

plasmid, which may act as a helper of its mobilization [35, 36] and should be of great concern.” **in the first version and** “On the other hand, although p1_Kpn_XM9 was not self-transmissible, it was mobilizable (Table 4) and Kpn_XM9 had a conjugative IncFII plasmid, which may act as a helper of its mobilization [40, 41] and should be of great concern.” **in the second version.**

Re: Spectrum00331-24R2 (Double *bla*_{KPC-2} copies quadrupled minimum inhibitory concentration of ceftazidime-avibactam in hospital-derived *Klebsiella pneumoniae*)

Dear Mr. Xiaobo Ma:

Your manuscript has been accepted, and I am forwarding it to the ASM production staff for publication. Your paper will first be checked to make sure all elements meet the technical requirements. ASM staff will contact you if anything needs to be revised before copyediting and production can begin. Otherwise, you will be notified when your proofs are ready to be viewed.

Sincerely,
Brian Conlon
Editor
Microbiology Spectrum